# Decentralizing Test-time Adaptation under Heterogeneous Data Streams

## Abstract

While Test-Time Adaptation (TTA) has shown promise in addressing distribution shifts between training and testing data, its effectiveness diminishes with heterogenous data streams due to uniform target estimation. As previous attempts merely stabilize model fine-tuning over time to handle continually changing environments, they fundamentally assume a homogeneous target domain at any moment, leaving the intrinsic real-world data heterogeneity unresolved. This paper delves into TTA under heterogeneous data streams, moving beyond current *model-centric* limitations. By revisiting TTA from a *data-centric* perspective, we discover that decomposing samples into Fourier space facilitates an accurate data separation across different frequency levels. Drawing from this insight, we propose a novel Frequency-based Decentralized Adaptation framework, which transitions data from globally heterogeneous to locally homogeneous in Fourier space and employs decentralized adaptation to manage diverse distribution shifts. Particularly, multiple local models are allowed to independently adjust to their specific data segments while periodically exchanging knowledge to form a cohesive global model. As such, not only can data diversity be captured, but also the overall model generalization can be enhanced across multiple distribution shifts. Importantly, we devise a novel Fourier-based augmentation strategy to assist in decentralizing adaptation, which selectively augments samples for each type of distribution shift and further enhances model robustness in complex real-world environments. Extensive experiments across various settings (corrupted, natural, and medical) demonstrate the superiority of our proposed framework over the state-of-the-arts.

## 1 Introduction

Deep learning models often suffer significant performance degradation when deployed in environments where the data distribution differs from that of the training set – a challenge known as domain shift (Long et al., 2013; Ganin & Lempitsky, 2015). Recently, Test-Time Adaptation (TTA) (Wang et al., 2021; Chen et al., 2022; Wang et al., 2022; Niu et al., 2022; 2023; Su et al., 2024; Press et al., 2024; Lee et al., 2024) has emerged as a promising solution by refining model parameters to better align with the encountered data at inference time. It leverages the incoming data stream for real-time adjustments without the need for retraining on a labeled dataset, enabling swift model adaptation to unpredictable data characteristics during deployment.

Despite their success, the effectiveness of current TTA models is generally constrained within ideal testing conditions – often involving homogeneous test samples with similar types of distribution shifts. While attempts have been made to address dynamic target distributions in continually changing environments (Wang et al., 2022), they fundamentally presume a uniform target domain at any time point. Their focus remains on enhancing model robustness against regular changes by stabilizing the fine-tuning process either by periodically resetting model weights (Niu et al., 2023; Press et al., 2024) or by down-weighting samples that deviate from the estimated distribution (Niu et al., 2022; Lee et al., 2024). Although these model-centric approaches may offer temporary relief, they do not fully recognize the intrinsic heterogeneity of real-world data. In practice, distribution shifts do not necessarily occur gradually over time but can be multifaceted at a single moment, involving heterogeneous and even conflicting shifts that current TTA models fail to adequately capture.

To address this, it is crucial to understand how these heterogeneous distribution shifts impact model adaptation. When a model attempts to adjust simultaneously to multiple diverse and potentially conflicting shifts, it may encounter adaptation conflicts. Specifically, adjustments made to accommodate one type of shift can interfere with adaptations for another, as different shifts may require conflicting changes to the model parameters. For instance, adapting to variations in image brightness might necessitate parameter updates that conflict with those needed for texture changes. Such conflicts prevent models from generalizing effectively across all encountered shifts, leading to irreversible degradation in predictive capabilities.

Recognizing these issues, we argue for *shifting from a model-centric to a data-centric approach that proactively addresses distribution diversity in Fourier space.* The rationale is that the frequency domain, unlike the common spatial domain, enables a clearer separation of data variations across different frequency levels. For example, high-frequency components are typically associated with fine-grained features like edges and textures, whereas low-frequency components generally relate to overall structural patterns such as shapes and illumination. By decomposing data into these frequency components, we can effectively isolate and manage different types of distribution shifts. Moreover, since the Fourier transform operates directly on the raw input images at the pixel level, it does not depend on pretrained model outputs, avoiding potential uncertainties due to significant distribution shifts. Importantly, this proactive separation allows us to manage distribution diversity prior to adaptation, offering a robust foundation for subsequent model enhancement.

Building on this insight, we introduce a framework termed Frequency-based Decentralized Adaptation (FreDA). Specifically, we first dynamically partition incoming data in the Fourier domain using high-frequency information. This initial segmentation facilitates the transition from globally heterogeneous to locally homogeneous data subsets before any model adaptation occurs. On this basis, we propose a decentralized learning strategy that allows multiple local models to independently adjust to their specific data segments while periodically exchanging knowledge to form a cohesive global model. This dual approach not only captures the diversity of distribution shifts to reduce potential conflicting adaptations but also leverages periodic communication among local models to enhance the global model's generalization across multiple shifts. Furthermore, we introduce a Fourier-based augmentation mechanism paired with an entropy-based sampling strategy, which significantly increases both the quantity and quality of samples for each type of shift. This enhancement further improves the model's robustness and predictive capabilities in dynamic environments. To summarize, the main contributions of this work are three-fold:

- We identify that many existing TTA methods are restricted in a *model-centric* paradigm that overlooks the data heterogeneity inherent in real-world scenarios. This oversight results in ineffective adaptation when facing diverse distribution shifts simultaneously.

- We revisit TTA from a *data-centric* perspective and introduce the FreDA framework. It reinterprets principles from both Fourier space and decentralized learning, leveraging specialized local adaptations to manage heterogeneous distribution shifts at test time.

- We conduct extensive evaluations of our method across a diverse range of datasets – including corrupted, natural, and medical scenarios – demonstrating its consistent superiority.

## 2 PRELIMINARIES

**Test-Time Adaptation under Mixed Distributions.** Test-time adaptation (TTA) aims to adjust a model $q_\theta(y|x)$, initially trained on a source dataset $\mathcal{D}_s = \{(x, y) \sim p_s(x, y)\}$, to a target domain $\mathcal{D}_t = \{(x, y) \sim p_t(x, y)\}$ without access to source data or target labels. Traditionally, TTA handles covariate shift by assuming $p_s(y|x) = p_t(y|x)$ while $p_s(x) \neq p_t(x)$. The challenge intensifies when $\mathcal{D}_t$ includes multiple non-i.i.d sub-distributions $p_{t,i}(x)$, complicating the adaptation process:

$$p_t(x) = \{p_{t,1}(x), p_{t,2}(x), \ldots, p_{t,N}(x)\}$$

This scenario requires the model $q_\theta(y|x)$ to effectively handle the heterogeneous and evolving target distribution to maintain robust performance. TTA strategies must therefore refine the model to optimize its predictive accuracy across these diverse sub-domains, ensuring consistent and reliable performance amidst significant distributional variability.

**Fourier Transformation.** Analyzing the frequency components of images is essential for understanding their underlying structures, and Fourier transformation plays a central role in this process. For a single-channel image $x$, its Fourier transformation $F(x)$ is given by: $F(x)(u,v) = \sum_{h=0}^{H-1} \sum_{w=0}^{W-1} x(h,w) e^{-j2\pi \left( \frac{h}{H} u + \frac{w}{W} v \right)}$ where $H$ and $W$ denote the height and width of the image, respectively, and $u$ and $v$ are the frequency coordinates. The inverse Fourier transformation $F^{-1}(x)$ allows for reconstructing the original image from its frequency spectrum, efficiently computed using the Fast Fourier Transform (FFT). In the frequency domain, images are characterized by amplitude $A(x)$ and phase $P(x)$ components, derived from the real $R(x)$ and imaginary $I(x)$ parts of $F(x)$:

$$A(x)(u,v) = \sqrt{R^2(x)(u,v) + I^2(x)(u,v)}, \quad P(x)(u,v) = \arctan \left( \frac{I(x)(u,v)}{R(x)(u,v)} \right), \quad (1)$$

where $A(x)$ reveals the intensity of the frequency content, e.g., high-frequency amplitudes highlight edges and fine details while low-frequency amplitudes emphasize the overall structure and gradual changes in the image, and $P(x)$ encodes the position of these features within the spatial domain.

## 3 CONNECTIONS TO PREVIOUS STUDIES

### 3.1 NON-I.I.D. IN TEST-TIME ADAPTATION

The non-i.i.d. problem in Test-Time Adaptation (TTA) challenges the conventional assumption that target batches are independent and identically distributed (i.i.d.), pushing the boundaries of TTA's applicability in real-world scenarios. This issue can be decomposed into two distinct challenges:

**Dependent Sampling.** This problem arises when the sampling within the target stream is dependent at the class level. Existing methods (Yuan et al., 2023; Gong et al., 2022; Zhao et al., 2023; Tomar et al., 2024; Marsden et al., 2024) have addressed this by aiming for class-balanced datasets during model updates, mitigating risks associated with class imbalance over time. They typically adjust sample proportions based on pseudo labels or extend data collection periods to reduce dependent sampling. However, unlike these methods that concentrate on mitigating class-level imbalances, our work focuses on enhancing TTA models in the presence of diverse sample styles or mixed distributions. We address data heterogeneity at the sample level, aiming to improve model adaptation capabilities in face of varying distribution shifts that are not captured by class balancing techniques. Notably, although our method is not tailored for class-dependent issues, our experimental results demonstrate that when class-dependent and mixed distributions coexist, our approach still achieves the best performance – showcasing the broad applicability of our model design.

**Mixed Distributions.** While attempts have been made to address dynamic target distributions in continually changing environments (Wang et al., 2022; Yuan et al., 2023; Niu et al., 2022; Press et al., 2024), they fundamentally assume a uniform target domain at each time point. Their approach focuses on strengthening model adaptation to constant changes by stabilizing the fine-tuning process, using periodic weight resets or down-weighting of unexpected samples. These *model-centric* approaches rely on uniform target estimation that fail to capture the actual data heterogeneity encountered in practice, causing model degradation in real-world deployment. In contrast, our work re-examines TTA from a *data-centric* perspective. We manage heterogeneous data streams by decomposing samples into the frequency domain, which facilitates an accurate data separation and allows us to address distribution diversity before adaptation occurs. Although a recent work (Niu et al., 2023) also consider mixed distribution scenarios, their study targets a broader "Dynamic Wild World" topic without delving deeply into this data heterogeneity problem. Conversely, our study focuses on managing heterogeneous data streams in TTA by leveraging the frequency domain and decentralized adaptation strategies to specifically address mixed data distributions at test time.

### 3.2 MULTI-TARGET DOMAIN ADAPTATION

Test-time Adaptation under mixed distribution resembles the multi-target unsupervised domain adaptation (MT-UDA) setting (Gholami et al., 2020; Isobe et al., 2021; Liu et al., 2020; Feng et al., 2024), where multiple domains exist within the target domain. However, TTA introduces complexities that far exceed those in conventional MT-UDA settings, primarily due to: **1)** Inaccessible

Labeled Source Data – In TTA, the labeled source distribution is not available, making it challenging to leverage source-target dissimilarities directly. **2) Dynamic and Unpredictable Target Streams** – TTA operates on a continuous influx of data, potentially incorporating new, unforeseen distributions, rather than a static, fully observable target dataset. This continuous nature of data flow prevents the establishment of a comprehensive understanding of the target distribution. These constraints complicates the formulation of adaptation strategies that depend on discerning the differences between the source and various subdomains within the target distribution.

### 3.3 DECENTRALIZED LEARNING, DISTRIBUTED LEARNING AND FEDERATED LEARNING

This work also intersects with decentralized, federated, and distributed learning due to our approach of splitting data batches into disjoint subsets and applying decentralized model adaptation: **1)** Decentralized learning typically focuses on learning from decentralized, non-i.i.d. data (Hsieh et al., 2020). In this work, however, the data is not originally decentralized; all target samples arrive together, while we proactively split them into disjoint subsets, revealing latent non-i.i.d. characteristics and enabling the effective use of decentralized learning techniques. **2)** Federated learning considers data privacy and multi-institutional collaborations within decentralized learning (McMahan et al., 2017). In our case, as target samples are mixed in a batch, data privacy is not a constraint. However, like federated learning, our approach also involves model collaboration where multiple local models periodically share insights to form a cohesive global model. **3)** Distributed learning aims to improve training efficiency on large-scale datasets by partitioning data for synchronized training (McDonald et al., 2010). In contrast, our method operates in a real-time fine-tuning context with limited data at one time, hence scalability is less of a concern.

### 3.4 FREQUENCY DOMAIN LEARNING

Frequency analysis has long been a cornerstone of conventional digital image processing. We focus on two key areas of Frequency Domain Learning that are particularly relevant to our topic:

**Frequency Information as a Tool for Analyzing DNN Behavior.** Research in deep learning has increasingly employed frequency analysis to uncover insights into Deep Neural Network (DNN) behavior, as highlighted in multiple studies (Wang et al., 2020; Xu, 2018; Xu et al., 2019; Yin et al., 2019). DNNs typically prioritize low-frequency features early in training, which represent the main structures of input data, aiding in stable and efficient learning. Conversely, high-frequency features, which detail finer, subtle variances, are crucial for improving a model's robustness to new or unseen domains. This understanding suggests that modulating the focus on different frequency bands during training can refine a model's performance across various conditions. By strategically enhancing the learning of high-frequency details, developers can better equip DNNs to handle diverse and challenging scenarios, balancing accuracy with domain adaptability.

**Frequency Information Enhances Model Adaptation and Generalization.** The utilization of frequency-based techniques, such as Fourier transforms, has become increasingly popular in transfer learning strategies. Within the Fourier spectrum, it's the phase component that mainly retains the high-level semantic content of signals, whereas the amplitude component generally encodes low-level statistical features. To capitalize on the ability of the Fourier phase to preserve semantic integrity, some methodologies (Yang & Soatto, 2020; Yang et al., 2022; Xu et al., 2021; 2023) incorporate a data augmentation process that involves linear interpolation between the amplitude spectra of different images. This approach effectively reduces the domain discrepancy in domain adaptation tasks and mitigates the risk of overfitting to the low-level statistical details present in the amplitude information, thus enhancing domain adaptation.

## 4 TTA UNDER MIXED DISTRIBUTION SHIFTS: A FOURIER PERSPECTIVE

### 4.1 MOTIVATIONS

Test-Time Adaptation (TTA) methods have been instrumental in managing domain shifts under a single type of target distribution. However, their effectiveness significantly diminishes under scenarios involving multiple distribution shifts. This is evident as models exhibit a marked decrease in sample class discriminability on the same dataset when exposed to mixed target distributions, as

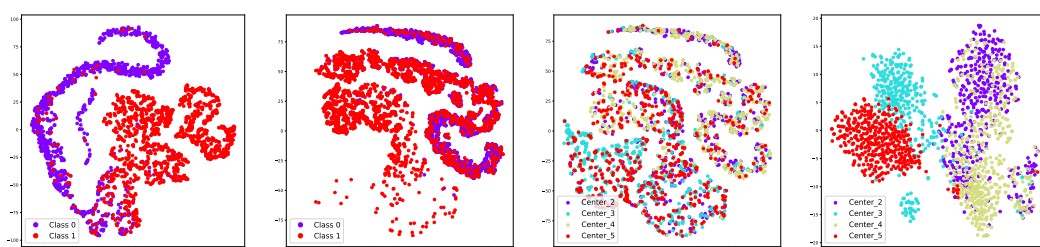

(a) Features produced by adapting models on single type of distribution shift where two classes are mostly separated.

(b) Features produced by adapting models on multiple types of distribution shift where two classes are largely overlapped.

(c) Features extracted by pretrained models from different centers are completely indistinguishable from one another.

(d) High-frequency information of images from four different centers facilitates clear and distinct separation among them.

Figure 1: t-SNE visualizations of the Camelyon17 dataset Bandi et al. (2018), including pathology slide images from five centers (domains), where the model is pretained on C1 and tested on C2-C5. (a-b) Features from C4 are presented as an example to illustrate the distinction in model adaptation when addressing single (a) vs multiple distribution shifts (b). (c-d) Comparison of features extracted by pretrained models and high-frequency information from images across four different domains.

demonstrated by comparing Figure 1(a) and (b). A direct approach to addressing this issue involves segregating samples belonging to different target distributions. However, in real-time applications, the specific target subdomains from which incoming samples originate, or whether they conform to the same distribution shifts, are generally agnostic. Attempting to cluster samples based on features extracted by the model can be misleading, as samples from different distributions may exhibit similarities due to belonging to the same category, resulting in poor separability of different target subdomains as shown in Figure 1(c). Interestingly, when clustering is based directly on the high-frequency information of the samples – without relying on model-derived feature extractors – a significant distinction can be made between samples from different target distributions, as shown in Figure 1(d). This observation is not unexpected, considering high-frequency information typically captures variations in image textures and styles, focusing more on the underlying differences in data distributions. Building on the experimental observations and analysis outlined above, the following section proposes leveraging the frequency domain for enhancing the adaptability of TTA methods in more realistic settings involving mixed distribution shifts.

## 4.2 Frequency-based Decentralized Adaptation

The previous discussions highlight how heterogeneity within target distribution can hinder model adaptation. This raises a natural question: *How can we manage this distributional heterogeneity to achieve better adaptation?* As established in our earlier section, effectively distinguishing samples associated with different distribution shifts is vital for successful domain adaptation. Moreover, the similarity in high-frequency information of samples provides a strong indication of whether they belong to the same or different target distributions.

Building upon these findings, we tackle the TTA problem by capitalizing on the high-frequency data components and propose a novel Frequency-based Decentralized Adaptation (FreDA) framework (see Figure 2). It employs a data-centric approach to partition target samples into multiple homogeneous subdomains in Fourier space, enabling an accurate model adaptation. This strategy is complemented by a novel frequency-based augmentation technique that enriches each target subdomain with synthetic samples, thereby further bolstering model adaptation. The overall pipeline of our proposed FreDA framework is detailed in Aglorithm 1.

### 4.2.1 Frequency-based Decentralized learning

**Insight:** Fourier transform offers an effective method to extract different frequency components from images, with high-frequency information particularly useful for capturing fine-grained details such as texture and noise. These details often highlight subtle variations among different distribution

shifts. By harnessing high-frequency components from images, we can distinguish samples that lead to different distribution shifts within a TTA setting through a simple clustering technique.

**Solution:** Based on this intuitive insight, we propose a new module called Frequency-based Decentralized Learning. This module leverages frequency information directly extracted from the pixel space to systematically partition data into multiple homogeneous subsets, enabling multiple local models to specialize in capturing each distribution shift individually. Concurrently, our method enhances collaborative learning by allowing periodic weight sharing among these local models, thereby boosting the overall model adaptability to diverse distribution shifts.

**Frequency Feature Extraction.** We start by extracting frequency domain features from the input images to identify distinct distribution shifts. Let $\mathbf{X} \in \mathbb{R}^{n \times c \times h \times w}$ denote a batch of input images, where $n$ is the batch size, $c$ is the number of channels, $h$ and $w$ are the height and width of the images. We first apply a Fourier transform $\mathcal{F}$ to each image $\mathbf{X}_i$ to obtain its frequency domain representation

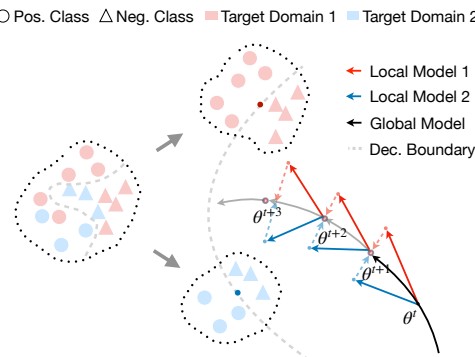

Figure 2: Decentralized adaptation simplifies class decision boundaries on heterogeneous data streams by enabling multiple local models to adapt towards the global optimum.

$\mathcal{F}(\mathbf{X}_i) \in \mathbb{C}^{h \times w \times c}$. Particularly, we focus on the amplitude spectrum $A(x)(u, v)$ in Eq. 1, filtering out low-frequency elements using mask $M(u, v) = \mathbb{1}\left(\left(u < \frac{h}{4} \vee u > \frac{3h}{4}\right) \vee \left(v < \frac{w}{4} \vee v > \frac{3w}{4}\right)\right)$ to emphasize the high-frequency components $G(x)(u, v)$ that are more likely to indicate shifts in distribution:

$$G(x)(u, v) = A(x)(u, v) \cdot M(u, v). \tag{2}$$

**Frequency-Based Clustering.** We then employ a clustering algorithm (e.g., K-means) to partition the frequency features into $K$ clusters, each corresponding to a different type of distribution shift. The clustering process is formalized as:

$$\min_{\mathbf{C}, \mathbf{Z}} \sum_{i=1}^{n} \|\mathbf{A}_{hf,i} - \mathbf{C}_{\mathbf{Z}_i}\|_2^2, \tag{3}$$

where $\mathbf{A}_{hf,i} = \text{vec}(G(x))$ represents the one-dimensional high-frequency component of the amplitude spectrum , $\mathbf{C} \in \mathbb{C}^{K \times hwc}$ represents the centroids of the clusters, and $\mathbf{Z} \in \{1, \ldots, K\}^n$ denotes the cluster assignments for each image.

**Decentralized Fine-tuning.** Test-time fine-tuning is then decentralized across these clusters, allowing for specialized adaptation within each subgroup: For each cluster $k$, we adapt a specialized model $q_{\theta_k}(y|x)$ that is fine-tuned using only the data within that cluster:

$$\theta_k^* = \arg\min_{\theta_k} \mathbb{E}_{x \sim p_{t,k}} \left[\mathcal{L}(q_{\theta_k}(x))\right], \tag{4}$$

where $p_{t,k}$ represents the data distribution within cluster $k$, and $\mathcal{L}$ is the loss function.

**Weight Aggregation.** To integrate knowledge from all subnetworks and prevent degradation on specific subdomains, we perform an aggregation of their parameters:

$$\mathbf{W}_{\text{global}} = \sum_{k=1}^{K} \left(\frac{|\mathcal{D}_k|}{\sum_{j=1}^{K} |\mathcal{D}_j|} \mathbf{W}_k\right), \tag{5}$$

where $|\mathcal{D}_k|$ denotes the number of samples in cluster $k$. This aggregation step combines the parameter updates from each subnetwork proportionally to its cluster size. The updated global model parameters $\mathbf{W}_{\text{global}}$ are then distributed back to each subnetwork, updating its parameters as follows:

$$\mathbf{W}_k \leftarrow \mathbf{W}_{\text{global}}. \tag{6}$$

---

**Algorithm 1** Framework of Frequency-based Decentralized Learning and Augmentation

---

**Require:** Step $t$, Input $\mathbf{X} \in \mathbb{R}^{n \times h \times w \times c}$, Pretrained source model $q_\theta$, Initialize Feature Repository $\mathcal{R} \leftarrow \emptyset$, CLUSTER_NUM $K$, KMEANS_SIZE $N$, COMM_INTERVAL $f$;

    **Step 1: Extract Frequency Features**

1: **for** $i = 1$ to $n$ **do**
2:     $\mathbf{A}_{hf,i} \leftarrow$ high_freq($\mathcal{F}(\mathbf{X}_i)$)             ▷ *Extract high-frequency components (Eq. 1, 2 )*
3: **end for**
    **Step 2: Dynamic Clustering**
4: $\mathcal{R} \leftarrow \mathcal{R} \cup \{\mathbf{A}_{hf,i}\}_{i=1}^n$                      ▷ *Frequency Information Repository*
5: $\mathcal{R} \leftarrow \mathcal{R}[(|\mathcal{R}| - N + 1) :]$            ▷ *Keep the last $N$ entries for kmeans clustering*
6: $(\mathbf{C}_t, \mathbf{Z}) \leftarrow$ K-means($\mathcal{R}, K, \mathbf{C}_{t-1}$)      ▷ *Obtain Cluster Labels $\mathbf{Z} = \{Z_i\}_{i=1}^n$ (Eq. 3)*
    **Step 3: Local Model Training**
7: **for** cluster $k \in \{1, \ldots, K\}$ **do**
8:     $\mathcal{S}_k \leftarrow \mathcal{S}_k \cup \{\mathbf{X}_i \mid Z_i = k\}$                ▷ *Gather samples for cluster $k$*
9:     $\mathcal{S}_k \leftarrow \mathcal{S}_k[(|\mathcal{S}_k| - n + 1) :]$        ▷ *Keep the last $batch\_size = n$ entries*
10:    $\mathcal{S}'_k \leftarrow$ select_samples($\mathcal{S}_k$)                ▷ *Select samples (Eq. 7)*
11:    **for** each $\mathbf{X}_i \in \mathcal{S}'_k$ **do**
12:       $\tilde{\mathbf{X}}_i \leftarrow$ augment($\mathbf{X}_i$)                  ▷ *Augment data (Eq. 9)*
13:       Train($q_{\theta_k}, \mathbf{X}_i, \tilde{\mathbf{X}}_i$)              ▷ *Train local model (Eq. 4)*
14:    **end for**
15: **end for**
    **Step 4: Compile Predictions**
16: $\mathbf{Y} \leftarrow$ collect_sort($\{q_{\theta_k}(\mathbf{X})\}$)            ▷ *Collect and sort predictions*
    **Step 5: Global Model Communication**
17: If t % $f == 0$ :             ▷ *Model Communication with interval $f$ (Eq.5, 6)*
18:    $\mathbf{W}_{\text{global}} \leftarrow \sum_{k=1}^K w_k \theta_k$
19:    $\mathbf{W}_k \leftarrow \mathbf{W}_{\text{global}}$

---

### 4.2.2 FREQUENCY-BASED AUGMENTATION

**Insight**: Although decentralized learning effectively handles data heterogeneity within the current batch and prevents the confusion of different distribution shifts, it may still suffer from inadequate characterization of each distribution shift due to limited batch data. Typically, TTA methods attempt to enhance the overall quality of observed target samples via data augmentation. However, traditional augmentation techniques in TTA, borrowed from standard computer vision practices such as rotation, clipping, and mixup, albeit beneficial in scenarios with single distribution shifts, struggle to guarantee targeted fine-tuning under more complex, mixed distribution shift scenarios.

**Solution:** To overcome these limitations, we propose a frequency-based augmentation strategy tailored for TTA under mixed distribution shifts. Unlike conventional techniques that apply general visual transformations, our method specifically perturbs the amplitue components of each sample in Fourier space. This targeted approach allows us to augment the target samples effectively within their respective distribution shifts, enhancing the quality of data available for each individual shifting case. By focusing on the frequency aspect, our strategy ensures that the model can generalize better by simulating and learning from an expanded range of each potential distribution shift, boosting the model's ability to adapt and perform robustly across varied scenarios.

**Sample Selection Mechanism.** Our sample selection mechanism leverages a criterion derived from the weighted entropy framework used in ETA (Niu et al., 2022) based on two primary conditions:

$$\text{Selection\_Criterion} = \mathbb{1}\left[(H(\mathbf{y}_t) < H_0) \wedge (|\cos(\mathbf{y}_t, \bar{\mathbf{y}}_{t-1})| < \epsilon)\right]. \tag{7}$$

The entropy $H(\mathbf{y}_t)$ measures the uncertainty in the current predictions. The cosine similarity $\cos(\mathbf{y}_t, \bar{\mathbf{y}}_{t-1})$ denotes the deviation between the current sample's class probabilities $\mathbf{y}_t$ and the aggregated class probabilities $\bar{\mathbf{y}}_{t-1}$. $\epsilon$ is the threshold for cosine similarity, and $H_0$ is the fixed entropy threshold. This ensures that selected samples exhibit significant deviations from previous predictions in class distribution and lower prediction uncertainty.

**Frequency-Based Augmentation.** The augmentation process involves perturbing the amplitude spectrum. Let $A(\mathbf{F}_i)$ represent the amplitude spectrum of a selected sample $\mathbf{X}_i$. To generate a

perturbed amplitude spectrum $\tilde{A}(\mathbf{F}_i)$, we apply a random Gaussian perturbation:

$$\tilde{A}(\mathbf{F}_i) = (1 + \alpha \cdot \Delta) \cdot A(\mathbf{F}_i), \tag{8}$$

where $\Delta \sim \mathcal{N}(0, \sigma^2)$ is a perturbation matrix sampled from a Gaussian distribution, and $\alpha$ is a scaling factor. Then, the synthetic sample $\tilde{\mathbf{X}}_i$ is reconstructed via the inverse Fourier transform to the perturbed amplitude spectrum, combined with the original phase spectrum $P(\mathbf{F}_i)$:

$$\tilde{\mathbf{X}}_i = \mathcal{F}^{-1}\left(\tilde{A}(\mathbf{F}_i), P(\mathbf{F}_i)\right). \tag{9}$$

**Loss Function.** The training objective in FreDA combines the entropy loss of the selected samples with a synthetic loss derived from the augmented samples. The total loss for a batch is defined as:

$$\mathcal{L}_{\text{total}} = \frac{1}{n}\sum_{i=1}^{n} H(\mathbf{y}_i) + \lambda \cdot \frac{1}{n}\sum_{i=1}^{n} \mathcal{L}_{\text{syn}}\left(\hat{\mathbf{y}}_i, \tilde{\mathbf{y}}_i\right), \tag{10}$$

where the entropy loss $H(\mathbf{y}_i)$ for the original samples is given by $H(\mathbf{y}_i) = -\sum_{j=1}^{C} \mathbf{y}_{i,j} \log \mathbf{y}_{i,j}$ with $\mathbf{y}_i$ being the predicted probability over the $C$ classes for the sample $\mathbf{X}_i$, and the synthetic loss $\mathcal{L}_{\text{syn}}\left(\tilde{\mathbf{y}}_i, \hat{\mathbf{y}}_i\right)$ is defined as the cross-entropy between the prediction $\tilde{\mathbf{y}}_i$ of the synthetic sample $\tilde{\mathbf{X}}_i$ and the pseudo-label $\hat{\mathbf{y}}_i$ from the original sample: $\mathcal{L}_{\text{syn}}\left(\hat{\mathbf{y}}_i, \tilde{\mathbf{y}}_i\right) = -\sum_{j=1}^{C} \hat{\mathbf{y}}_{i,j} \log \tilde{\mathbf{y}}_{i,j}$.

## 5 EXPERIMENTS

**Datasets and Experimental Settings.** To provide a comprehensive evaluation of TTA deployment, we test models over multiple datasets under three different scenarios:

- Common Image Corruptions: We evaluate models on CIFAR-10-C, CIFAR-100-C, and ImageNet-C (Hendrycks & Dietterich, 2018) with 10, 100 and 1000 classes, respectively. These benchmarks are designed to assess the robustness of classification networks against various corruptions. Each dataset consists of images subjected to 15 distinct corruptions across five severity levels, resulting in 150,000 data at each severity for CIFAR-10-C and CIFAR-100-C, and 750,000 for ImageNet-C.
- Natural Domain Shifts: We extend our evaluation to DomainNet126 (Saito et al., 2019), which presents natural shifts across four domains (Real, Clipart, Painting, Sketch) encompassing 126 classes, representing a subset of the larger DomainNet dataset.
- Medical Application: Models are further evaluated on Camelyon17 (Bandi et al., 2018), comprising over 450,000 histopathological patches from lymph node sections for binary classification of normal and tumor tissue, with data originating from five distinct healthcare centers.

For corruption datasets, the model is pretrained on the clean dataset and the 15 corruptions are randomly mixed as the target distribution. In DomainNet126 and Camelyon17, one subdomain is selected as the source, and the others serve as mixed target distributions. More implementation details are provided in Appendix A.

**Adaptation Scenarios.** To evaluate models in adapting to heterogeneous data streams, we focus on two primary distribution shift scenarios including:

- Mixed Domains: The model adapts to a long test sequence where consecutive test samples may come from different domains.
- Mixed Domains & Dependent Sampling: This scenario extends the mixed distribution framework by introducing sequential, time-correlated data from the same class across ordered domains, featuring both covariate and label shifts.

While our primary focus is on mixed domains, we have also included the commonly used continual setting for evaluation. Due to space limit, detailed experimental results are provided in Appendix C.

**Baselines.** We compare our FreDA with 10 models: TBN Nado et al. (2020), TENT (Wang et al., 2021), CoTTA (Wang et al., 2022), ETA (Niu et al., 2022), SAR (Niu et al., 2023), AdaContrast (Chen et al., 2022), RoTTA (Yuan et al., 2023), RDumb (Press et al., 2024), DeYO (Lee et al., 2024), and UnMix-TNS (Tomar et al., 2024). See more information in Appendix B.

Table 1: Classification error rate (↓) on CIFAR-10-C, CIFAR-100-C, and ImageNet-C (IN-C) respectively using WRN-28, ResNeXt-29, ResNet-50-BN and VitBase-LN backbones under **Mixed Distribution**. The corruption severity is 5 and the result is averaged over three runs.

| Baseline & Methods | Gauss. | Shot | Impul. | Defoc. | Glass | Motion | Zoom | Snow | Frost | Fog | Brig. | Contr. | Elast. | Pixel | JPEG | **Avg.** |
|---|---|---|---|---|---|---|---|---|---|---|---|---|---|---|---|---|
| **CIFAR-10-C (WRN-28)** | 72.3 | 65.7 | 72.9 | 46.9 | 54.3 | 34.8 | 42.0 | 25.1 | 41.3 | 26.0 | 9.3 | 46.7 | 26.6 | 58.4 | 30.3 | 43.5 |
| TBN | 45.5 | 42.8 | 59.7 | 34.2 | 44.3 | 29.8 | 32.0 | 19.8 | 21.1 | 21.5 | 9.3 | 27.9 | 33.1 | 55.5 | 30.8 | 33.8 |
| TENT (ICLR 21') | 73.5 | 70.1 | 81.4 | 31.6 | 60.3 | 29.6 | 28.5 | 30.8 | 35.3 | 25.7 | 13.6 | 44.2 | 32.6 | 70.2 | 34.9 | 44.1 |
| ETA (ICML 22') | 36.2 | 33.3 | 52.3 | 22.9 | 38.9 | 22.4 | 20.5 | 19.5 | 19.7 | 20.4 | 11.3 | 35.4 | 26.6 | 38.8 | 25.1 | 28.2 |
| AdaContrast (CVPR 22') | 36.7 | 34.3 | 48.8 | 18.2 | 39.1 | 21.1 | 17.7 | 18.6 | 18.3 | 16.8 | 9.0 | 17.4 | 27.7 | 44.8 | 24.9 | 26.2 |
| CoTTA (CVPR 22') | 38.7 | 36.0 | 56.1 | 36.0 | 36.8 | 32.3 | 31.0 | 19.9 | 17.6 | 27.2 | 11.7 | 52.6 | 30.5 | 35.8 | 25.7 | 32.5 |
| SAR (ICLR 23') | 45.5 | 42.7 | 59.6 | 34.1 | 44.3 | 29.7 | 31.9 | 19.8 | 21.1 | 21.5 | 9.3 | 27.8 | 33.0 | 55.4 | 30.8 | 33.8 |
| RoTTA (CVPR 23') | 60.0 | 55.5 | 70.0 | 23.8 | 44.1 | 20.7 | 21.3 | 20.2 | 22.7 | 16.0 | 9.4 | 22.7 | 27.0 | 58.6 | 29.2 | 33.4 |
| RDumb (NeurIPS 23') | 34.9 | 32.3 | 49.4 | 23.3 | 38.2 | 23.3 | 20.7 | 19.9 | 19.3 | 20.7 | 11.2 | 29.3 | 26.7 | 41.5 | 25.2 | 27.7 |
| DeYO (ICLR 24') | 45.8 | 42.3 | 65.7 | 21.3 | 41.8 | 25.1 | 19.5 | 21.1 | 19.6 | 19.2 | 12.3 | 21.8 | 28.5 | 39.3 | 28.0 | 30.1 |
| UnMix-TNS (ICLR 24') | 50.0 | 44.4 | 44.3 | 34.4 | 48.2 | 32.7 | 30.0 | 35.5 | 35.9 | 47.5 | 28.1 | 38.7 | 43.9 | 40.0 | 43.3 | 39.8 |
| **FreDA (ours)** | 23.1 | 22.2 | 32.2 | 18.7 | 41.6 | 18.8 | 16.8 | 17.9 | 19.9 | 16.9 | 9.8 | 13.2 | 29.1 | 35.4 | 28.6 | 22.9 |
| **CIFAR-100-C (ResNeXt-29)** | 73.0 | 68.0 | 39.4 | 29.3 | 54.1 | 30.8 | 28.8 | 39.5 | 45.8 | 50.3 | 29.5 | 55.1 | 37.2 | 74.7 | 41.2 | 46.4 |
| TBN | 62.7 | 60.7 | 43.1 | 35.5 | 50.3 | 35.7 | 34.4 | 39.9 | 51.5 | 27.5 | 45.5 | 42.3 | 72.8 | 46.4 | 45.8 | 45.8 |
| TENT (ICLR 21') | 95.6 | 95.2 | 89.2 | 72.8 | 82.9 | 74.4 | 72.3 | 78.0 | 79.7 | 84.7 | 71.0 | 88.5 | 77.8 | 96.8 | 78.7 | 82.5 |
| ETA (ICML 22') | 42.6 | 40.3 | 34.1 | 30.3 | 42.4 | 32.0 | 29.4 | 35.6 | 35.8 | 44.1 | 30.2 | 41.8 | 36.9 | 38.9 | 40.9 | 37.0 |
| AdaContrast (CVPR 22') | 54.5 | 51.5 | 37.6 | 30.7 | 45.4 | 32.1 | 30.3 | 36.9 | 36.5 | 45.3 | 28.0 | 42.7 | 38.2 | 75.4 | 41.7 | 41.8 |
| CoTTA (CVPR 22') | 54.4 | 52.7 | 49.8 | 36.0 | 45.8 | 36.7 | 33.9 | 38.9 | 35.8 | 52.0 | 30.4 | 60.9 | 40.2 | 38.0 | 41.1 | 43.1 |
| SAR (ICLR 23') | 75.8 | 72.7 | 41.1 | 29.2 | 50.0 | 31.1 | 28.9 | 36.7 | 37.7 | 43.9 | 29.3 | 41.8 | 37.1 | 89.2 | 42.4 | 45.5 |
| RoTTA (CVPR 23') | 65.0 | 62.3 | 39.3 | 33.4 | 50.0 | 34.2 | 32.6 | 36.6 | 36.5 | 45.0 | 26.4 | 41.6 | 40.6 | 89.5 | 48.5 | 45.4 |
| RDumb (NeurIPS 23') | 42.3 | 40.0 | 34.1 | 30.5 | 42.4 | 31.9 | 29.5 | 35.7 | 35.9 | 43.6 | 30.4 | 41.9 | 36.9 | 38.1 | 40.5 | 36.9 |
| DeYO (ICLR 24') | 57.2 | 53.4 | 38.8 | 34.7 | 47.3 | 37.3 | 34.1 | 40.8 | 40.5 | 50.6 | 33.3 | 45.8 | 41.5 | 94.5 | 45.7 | 46.4 |
| UnMix-TNS (ICLR 24') | 65.8 | 64.1 | 46.4 | 37.5 | 51.7 | 36.0 | 36.4 | 38.5 | 39.4 | 51.1 | 29.3 | 42.8 | 43.2 | 67.8 | 49.4 | 46.6 |
| **FreDA (ours)** | 34.8 | 34.7 | 36.6 | 29.4 | 41.2 | 29.9 | 28.4 | 33.8 | 33.7 | 41.1 | 29.8 | 34.9 | 36.2 | 36.9 | 37.1 | 34.7 |
| **IN-C (ResNet-50-BN)** | 97.8 | 97.1 | 98.2 | 81.7 | 89.8 | 85.2 | 77.9 | 83.5 | 77.1 | 75.9 | 41.3 | 94.5 | 82.5 | 79.3 | 68.6 | 82.0 |
| TBN | 92.8 | 91.1 | 92.5 | 87.8 | 90.2 | 87.2 | 82.2 | 82.2 | 82.0 | 79.8 | 48.0 | 92.5 | 83.5 | 75.6 | 70.4 | 82.5 |
| TENT (ICLR 21') | 99.2 | 98.7 | 99.0 | 90.5 | 95.1 | 90.5 | 84.6 | 86.6 | 84.0 | 86.5 | 46.7 | 98.1 | 86.1 | 77.7 | 72.9 | 86.4 |
| ETA (ICML 22') | 90.7 | 89.2 | 90.5 | 77.0 | 80.6 | 74.0 | 68.9 | 72.4 | 70.3 | 64.6 | 43.9 | 93.4 | 69.2 | 52.3 | 55.9 | 72.9 |
| AdaContrast (CVPR 22') | 96.2 | 95.5 | 96.2 | 93.2 | 96.4 | 96.3 | 90.5 | 92.7 | 91.9 | 92.4 | 50.8 | 97.0 | 96.6 | 89.7 | 87.1 | 90.8 |
| CoTTA (CVPR 22') | 89.1 | 86.6 | 88.5 | 80.9 | 87.2 | 81.1 | 75.8 | 73.3 | 73.2 | 70.5 | 41.6 | 85.0 | 78.1 | 65.6 | 61.6 | 76.0 |
| SAR (ICLR 23') | 98.4 | 97.3 | 98.0 | 84.0 | 87.3 | 82.6 | 77.2 | 77.5 | 76.1 | 72.5 | 43.1 | 96.0 | 78.3 | 61.8 | 60.4 | 79.4 |
| RoTTA (CVPR 23') | 89.4 | 88.6 | 89.3 | 83.4 | 89.1 | 86.2 | 80.0 | 78.9 | 76.9 | 74.2 | 37.4 | 89.6 | 79.5 | 69.0 | 59.6 | 78.1 |
| RDumb (NeurIPS 23') | 89.0 | 87.6 | 88.6 | 78.1 | 82.3 | 75.2 | 70.1 | 73.0 | 71.0 | 65.1 | 43.9 | 92.6 | 70.7 | 53.7 | 56.3 | 73.1 |
| DeYO (ICLR 24') | 99.5 | 99.2 | 99.5 | 89.5 | 95.0 | 83.9 | 78.8 | 75.0 | 87.8 | 79.2 | 47.3 | 99.2 | 92.4 | 59.0 | 60.4 | 83.0 |
| UnMix-TNS (ICLR 24') | 91.7 | 92.8 | 91.7 | 92.3 | 93.4 | 84.8 | 84.8 | 86.3 | 86.3 | 84.1 | 85.0 | 62.0 | 96.5 | 88.6 | 81.7 | 86.7 |
| **FreDA (ours)** | 72.4 | 74.0 | 71.4 | 76.5 | 82.3 | 72.1 | 64.1 | 64.4 | 64.8 | 59.1 | 43.7 | 79.7 | 71.0 | 54.2 | 58.6 | 67.2 |
| **IN-C (VitBase-LN)** | 65.8 | 67.3 | 65.3 | 68.8 | 74.4 | 64.3 | 66.6 | 56.8 | 45.2 | 48.6 | 29.2 | 81.8 | 57.1 | 60.8 | 50.2 | 60.2 |
| TENT (ICLR 21') | 60.6 | 60.4 | 59.6 | 63.6 | 67.8 | 57.1 | 62.1 | 55.0 | 48.8 | 47.4 | 28.6 | 66.7 | 53.9 | 50.4 | 44.4 | 55.0 |
| ETA (ICML 22') | 59.3 | 57.8 | 57.9 | 58.8 | 62.8 | 52.5 | 58.2 | 51.0 | 46.4 | 44.2 | 28.8 | 58.3 | 51.1 | 46.9 | 41.9 | 51.7 |
| AdaContrast (CVPR 22') | 64.8 | 63.4 | 63.3 | 72.8 | 76.6 | 73.7 | 74.6 | 67.7 | 48.0 | 89.6 | 30.2 | 93.2 | 60.8 | 57.3 | 46.3 | 65.5 |
| CoTTA (CVPR 22') | 89.4 | 92.0 | 88.9 | 93.6 | 92.6 | 90.6 | 86.5 | 94.9 | 88.2 | 86.6 | 75.8 | 96.5 | 85.7 | 93.5 | 84.6 | 89.3 |
| SAR (ICLR 23') | 58.9 | 57.6 | 57.6 | 59.4 | 63.6 | 53.0 | 58.5 | 52.3 | 47.1 | 45.4 | 28.3 | 61.6 | 51.4 | 47.4 | 42.0 | 52.3 |
| RoTTA (CVPR 23') | 64.4 | 65.6 | 63.7 | 67.6 | 71.3 | 59.8 | 64.1 | 52.7 | 43.5 | 48.6 | 27.9 | 54.7 | 54.3 | 60.4 | 50.1 | 58.2 |
| RDumb (NeurIPS 23') | 59.7 | 58.5 | 58.5 | 60.0 | 64.1 | 54.0 | 59.0 | 52.0 | 46.7 | 44.5 | 28.6 | 61.2 | 51.9 | 48.3 | 42.6 | 52.6 |
| DeYO (ICLR 24') | 60.0 | 58.6 | 58.8 | 58.8 | 62.4 | 61.9 | 50.9 | 46.7 | 51.9 | 45.2 | 29.7 | 55.7 | 51.6 | 45.8 | 42.8 | 52.1 |
| **FreDA (ours)** | 55.9 | 53.7 | 55.0 | 58.0 | 57.9 | 50.9 | 57.4 | 45.5 | 42.9 | 43.9 | 29.5 | 51.7 | 47.8 | 41.6 | 40.7 | 48.8 |

**FreDA consistently improves across different distribution shifts.** Our method consistently attains the lowest classification error rates across all evaluated datasets (see Table 1 and 3). Notably, on the Camelyon17 dataset, FreDA reduced the error rate to 27.9%, outperforming the next best method by 5.9%. This significant improvement is particularly notable where other approaches falter – especially compared to models like TBN without training, which struggle to adapt to the complex medical imaging data. By effectively handling high variability and intricate patterns in the data, FreDA maintains superior accuracy and adaptability. These results demonstrate our method's practical utility in applications where effective adaptation to new and unseen conditions is essential, underscoring its robustness and reliability for real-world deployment.

Table 2: Classification error rate (↓) on CIFAR-10-C (C10), CIFAR-100-C (C100), and ImageNet-C (IN) using ResNet-50-BN and VitBase-LN backbones under **Mixed Distribution & Dependent Sampling**, averaged over 15 corruptions at severity level 5.

| Methods | C10 | C100 | IN(BN) | IN(LN) |
|---|---|---|---|---|
| Source | 43.5 | 46.5 | 82.0 | 60.2 |
| TBN | 79.2 | 92.3 | 94.2 | - |
| TENT (ICLR 21') | 86.6 | 98.4 | 99.5 | 77.9 |
| ETA (ICML 22') | 86.1 | 96.2 | 99.7 | 73.9 |
| AdaContrast (CVPR 22') | 69.8 | 73.2 | 98.5 | 94.9 |
| CoTTA (CVPR 22') | 82.7 | 92.8 | 98.0 | 92.6 |
| SAR (ICLR 23') | 78.8 | 95.8 | 98.2 | 54.0 |
| RoTTA (CVPR 23') | 64.6 | 65.3 | 89.3 | 74.2 |
| RDumb (NeurIPS 23') | 86.2 | 98.4 | 98.1 | 56.5 |
| DeYO (ICLR 24') | 87.0 | 98.1 | 99.1 | 52.0 |
| UnMix-TNS (ICLR 24') | 41.9 | 50.1 | 84.3 | - |
| **FreDA (ours)** | 23.0 | 34.7 | 67.2 | 48.7 |

**FreDA effectively mitigates both covariate and label shifts.** In environments characterized by simultaneous covariate and label shifts, our approach keep showing exceptional adaptability (see Table 2). We attribute this success to FreDA's ability to separate covariate shifts from label shifts via decentralized learning. FreDA achieves this by first isolating target different distribution shifts and then focus on learning label shifts for each specific distribution. This sequential approach prevents models from being overwhelmed by simultaneous shifts, allowing it to address each type of shift independently and effectively.

Table 3: Classification error rate (↓) on DomainNet126 and Camelyon17 under **Mixed Distribution**.

| | DomainNet126 | | | | | Camelyon17 | | | | | |
|---|---|---|---|---|---|---|---|---|---|---|---|
| Methods | Real | Painting | Clipart | Sketch | Avg. | A | B | C | D | E | Avg. |
| Source | 45.2 | 41.6 | 49.5 | 45.3 | 45.4 | 21.6 | 43.6 | 52.5 | 47.4 | 47.6 | 42.5 |
| TBN | 45.5 | 39.9 | 45.9 | 37.5 | 42.2 | 26.5 | 38.5 | 31.7 | 39.4 | 32.8 | 33.8 |
| TENT (ICLR 21') | 42.2 | 37.8 | 44.7 | 37.5 | 40.6 | 44.7 | 50.5 | 49.9 | 49.1 | 48.6 | 48.6 |
| ETA (ICML 22') | 41.1 | 37.3 | 43.4 | 36.4 | 39.5 | 47.4 | 52.5 | 47.9 | 49.9 | 39.2 | 47.4 |
| SAR (ICLR 23') | 43.2 | 38.5 | 44.8 | 37.0 | 40.9 | 26.5 | 38.5 | 31.7 | 39.4 | | 33.8 |
| DeYO (ICLR 24') | 40.9 | 36.4 | 43.6 | 36.9 | 39.4 | 50.4 | 50.3 | 48.8 | 51.7 | 50.5 | 50.4 |
| **FreDA (ours)** | **40.2** | **36.1** | **40.0** | **33.6** | **37.5** | **18.6** | **24.7** | **24.8** | 40.5 | **30.8** | **27.9** |

Table 4: Classification error rate (↓) on CIFAR-10-C , CIFAR-100-C , and ImageNet-C using WRN-28, ResNeXt-29 and ResNet-50-BN backbones with **Various Batch Size (BS) under Mixed Domains**, averaged over 15 corruptions at severity level 5.

| | CIFAR-10-C | | | | | | CIFAR-100-C | | | | | | ImageNet | | | | |
|---|---|---|---|---|---|---|---|---|---|---|---|---|---|---|---|---|---|
| Methods | BS=200 | BS=64 | BS=16 | BS=4 | BS=1 | Avg. | BS=200 | BS=64 | BS=16 | BS=4 | BS=1 | Avg. | BS=64 | BS=16 | BS=4 | BS=1 | Avg. |
| Source | 43.5 | 43.5 | 43.5 | 43.5 | 43.5 | 43.5 | 46.4 | 46.4 | 46.4 | 46.4 | 46.4 | 46.4 | 82.0 | 82.0 | 82.0 | 82.0 | 82.0 |
| TBN | 33.8 | 34.1 | 35.5 | 40.7 | 89.8 | 46.8 | 45.8 | 46.5 | 49.5 | 60.4 | 98.9 | 60.2 | 82.5 | 83.7 | 89.0 | 99.9 | 88.8 |
| TENT (ICLR 21') | 44.1 | 57.1 | 80.3 | 88.4 | 90.0 | 72.0 | 82.5 | 92.2 | 97.1 | 98.6 | 99.0 | 93.9 | 86.4 | 99.6 | 99.8 | 99.9 | 96.4 |
| ETA (ICML 22') | 28.2 | 34.8 | 55.1 | 69.6 | 89.8 | 55.5 | 37.0 | 40.8 | 53.5 | 93.2 | 98.9 | 64.7 | 72.9 | 99.5 | 99.3 | 99.9 | 92.9 |
| SAR (ICLR 23') | 33.8 | 33.7 | 35.4 | 41.1 | 89.8 | 46.8 | 45.5 | 57.2 | 67.4 | 69.0 | 98.9 | 67.6 | 79.4 | 89.0 | 87.8 | 99.9 | 89.0 |
| DeYO (ICLR 24') | 27.7 | 34.4 | 44.8 | 78.9 | 89.8 | 55.1 | 46.4 | 66.4 | 95.0 | 98.3 | 98.9 | 81.0 | 83.0 | 97.0 | 87.8 | 99.9 | 91.9 |
| **FreDA (ours)** | **22.9** | **22.9** | **22.4** | **22.6** | **22.9** | **22.7** | **34.7** | **34.5** | **35.0** | **34.6** | **36.4** | **35.0** | **67.2** | **67.9** | **69.3** | **70.7** | **68.8** |

**FreDA remains stable under various batch size.** To simulate real-world deployment with constrained batch sizes, we evaluate models under both varying batch sizes and mixed distribution shifts. In Table 4, we present classification error rates on CIFAR-10-C, CIFAR-100-C, and ImageNet-C datasets using batch sizes ranging from 200 (64) down to 1. Unlike other methods that significantly degrade as batch size decreases – for example the error rate of DeYO increases from 27.7% to 89.8% when batch size drops from 200 to 1 on CIFAR-10-C – FreDA consistently maintains strong performance. This stability under limited batch sizes demonstrates FreDA's robustness, making it highly suitable for real-world applications where processing large batches is not always feasible.

**FreDA enhances adaptation via synergistic designs.** This section validates our designs by ablating its three key modules – Decentralized Training (DT), Sample Selection (SS), and Sample Augmentation (SA). The baseline here leverages only the entropy loss. From Table 5, we have the following observations: **1)** Implementing decentralized training alone results in substantial improvements, reducing error rates dramatically across all datasets. **2)** The impact of sample selection varies across datasets. While significantly improving performance on CIFAR100-C, it increase error rate on Camelyon. This

Table 5: Ablation study of FreDA.

| DT | SS | SA | C10 | C100 | IN(BN) | IN(LN) |
|---|---|---|---|---|---|---|
| | | | 44.1 | 82.5 | 86.4 | 55.0 |
| ✓ | | | 24.8 | 54.2 | 81.2 | 95.2 |
| | ✓ | | 29.6 | 37.5 | 71.0 | 51.1 |
| | | ✓ | 39.4 | 71.7 | 92.9 | 59.5 |
| ✓ | ✓ | | 24.3 | 36.3 | 69.4 | 49.6 |
| | ✓ | ✓ | 27.7 | 36.2 | 65.9 | 50.1 |
| ✓ | | ✓ | 24.4 | 50.2 | 77.7 | 95.3 |
| ✓ | ✓ | ✓ | 22.9 | 34.7 | 67.2 | 48.8 |

variation suggests that sample selection helps the model focus on more representative or challenging samples but may not be effective across all datasets, highlighting its dataset-specific nature. **3)** Sample augmentation alone tends to increase error rates, suggesting that although this approach introduces useful variability, it may introduce unexpected noise under the absence of proper selection or decentralized training. **4)** The combined approach delivers the best performance across all datasets, showing the synergistic effect of our different designs.

## 6 CONCLUSION

This paper advances Test-Time Adaptation (TTA) by addressing the real-world complexities of heterogeneous data streams. Our decentralized approach, leveraging Fourier information, enables a precise management of diverse data shifts, enhancing model adaptability and robustness across different settings. The integration of Fourier-based augmentation broadens the effective range of confident samples tailored for each distinct distribution shifts, leading to notable performance gains on multiple dataset across various domains. The demonstrated improvements confirm the potential of our proposed FreDA to significantly impact the field, suggesting promising avenues for future research in adapting to dynamic and diverse distributional changes in deep learning applications.

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

# A    IMPLEMENTATION DETAILS

**Pretrained Models.**    We utilize models from RobustBench (Croce et al., 2021), including WildResNet-28 (Zagoruyko & Komodakis, 2016) for CIFAR-10-C and ResNeXt-29 (Xie et al., 2017) for CIFAR-100-C, both pretrained by Hendrycks et al. (2020). For ImageNet-C, the pretrained ResNet-50 (He et al., 2016) and VitBase-LN Dosovitskiy (2020) are obtained from `torchvision`. For DomainNet126, pretrained ResNet-50 are sourced from AdaContrast (Chen et al., 2022), while for Camelyon17, we train a DenseNet-121 (Huang et al., 2017) from scratch to 100 epochs with other training specifications outlined in the Wilds benchmark (Koh et al., 2021).

**Hyperparameter Configuration.** The batch size is set to 200, 64, 128 and 32 for CIFAR-10/100-C, ImageNet-C, DomainNet126 and Camelyon17 following the previous methods. The SGD optimizer is used with learning rates adjusted to 0.01, 0.0001, 0.001 and 0.00005, respectively. The learning rate is proportionally decreased in the experiment studying the effect of batch size. The Kmeans Size is 512, Clutser Number is 4, Communication Interval is 10 across all the tasks. The perutrbation magnitude $\alpha$ is fixed to 0.1 and the coefficient $\lambda$ in loss function is fixed to 0.5. The $\delta$ parameter controlling the dependent sampling (Dirichlet distribution) is set to 0.1 for CIFAR10-C and adjusted to 0.01 for CIFAR100-C, ImageNet-C following UnMix-TNS (Tomar et al., 2024). Two threshold in Eq. 7 is set to the same value for corruption datasets and DomainNet126 following ETA (Niu et al., 2022). While for Camelyon17, the class diversity related threshold is adjusted to 0.9 empirically.

# B    COMPARED METHODS

TBN Nado et al. (2020) re-estimates batch normalization statistics from test data. TENT (Wang et al., 2021) minimizes prediction entropy to optimize batch normalization. CoTTA (Wang et al., 2022) addresses long-term test-time adaptation in changing environments. ETA (Niu et al., 2022) and SAR (Niu et al., 2023) exclude unreliable and redundant samples during optimization. Ada-Contrast (Chen et al., 2022) utilizes contrastive learning to refine pseudo-labels and improve feature learning. RoTTA (Yuan et al., 2023) presents a robust batch normalization scheme with a memory bank for category-balanced estimation. RDumb (Press et al., 2024) leverages weighted entropy and periodically resets the model to its pretrained state to prevent collapse. DeYO (Lee et al., 2024) quantifies the impact of object-destructive transformations for sample selection and weighting. UnMix-TNS (Tomar et al., 2024) introduces a test-time normalization layer for non-i.i.d. environments by decomposing BN statistics. For fair comparisons, we conduct experiments using the open source online TTA repository (Döbler et al., 2023)[1], which provides codes and configurations of state-of-the-art TTA methods.

# C    CONTINUAL SETTING EVALUATION

Although our method is specifically designed for mixed domain scenarios, we also evaluated its performance under the conventional continual test-time adaptation setting to assess its robustness in different contexts. In this setting, the model adapts online to a sequence of test domains without explicit knowledge of domain shifts, with only one distribution shift occurring at a time and not reappearing. Without adjusting any parameters, our method demonstrated competitive performance compared to current state-of-the-art approaches. Notably, while UnMix-TNS effectively addresses non-i.i.d. issues (dependent sampling at the class level) but underperforms under i.i.d. conditions, our results suggest that the proposed FreDA not only excels in its intended mixed domain scenarios but also generalizes effectively to standard continual adaptation tasks, providing a robust solution across various distributional challenges.

Table 6: Classification error rate ($\downarrow$) on CIFAR-10-C (C10), CIFAR-100-C (C100), and ImageNet-C (IN) using ResNet-50-BN & VitBase-LN backbones under **Continual Setting**, averaged over 15 corruptions.

| Methods | C10 | C100 | IN(BN) | IN(LN) |
|---|---|---|---|---|
| Source | 43.5 | 46.5 | 82.0 | 60.2 |
| TBN | 20.4 | 35.4 | 68.6 | - |
| TENT (ICLR 21') | 20.0 | 62.2 | 62.6 | 54.5 |
| ETA (ICML 22') | 17.9 | _32.2_ | 60.2 | _49.8_ |
| AdaContrast (CVPR 22') | 18.5 | 33.5 | 65.5 | 57.0 |
| CoTTA (CVPR 22') | **16.5** | 32.8 | 63.1 | 77.0 |
| SAR (ICLR 23') | 20.4 | **32.0** | _61.9_ | 51.7 |
| RoTTA (CVPR 23') | 19.3 | 34.8 | 67.3 | 58.3 |
| RDumb (NeurIPS 23') | _17.8_ | 34.1 | 90.6 | 50.2 |
| DeYO (ICLR 24') | 87.0 | 98.1 | 90.6 | 94.3 |
| UnMix-TNS (ICLR 24') | 24.9 | 32.7 | 75.4 | - |
| **FreDA (ours)** | 19.5 | 32.5 | **60.2** | **47.9** |

---

[1] https://github.com/mariodoebler/test-time-adaptation

## D PARAMETER STUDY

In this section, we study the parameter choice of CLUSTER_NUM, KMEANS_SIZE and COMM_INTERVAL (refer to Algorithm 1 for detailed definitions). Results are reported in Table 7.

As we adjust the KMEANS_SIZE parameter from 256 to 2048, there is a remarkably consistent performance on different datasets, indicating that our method's capability to generalize across various sizes.

The variation in CLUSTER_NUM across our datasets underscores the nuanced balance required in selecting the optimal branch number for domain adaptation. Utilizing just two clusters already yields relatively good results, suggesting that a minimal decentralization can be effective. However, as the number of clusters increases from 2 to 16, we observe a decline in performance on CIFAR100-C and a more pronounced deterioration on ImageNet-C, with the optimal performance achieved at a CLUSTER_NUM of 4. This trend underscores the delicate trade-off between model complexity and the risk of overfitting: employing too large a cluster size can lead to a model overly tailored to the training data, impairing its generalization capabilities.

Table 7: Sensitivity analysis on different datasets.

| CLUSTER_NUM | 2 | **4** | 8 | 16 |
| --- | --- | --- | --- | --- |
| CIFAR10-C | 23.0 | 22.9 | 23.2 | 24.7 |
| CIFAR100-C | 34.8 | 34.7 | 34.7 | 35.6 |
| IN-C (BN) | 68.6 | 67.2 | 67.1 | 70.5 |
| IN-C (LN) | 50.3 | 48.8 | 49.9 | 50.0 |
| **KMEANS_SIZE** | 256 | **512** | 1024 | 2048 |
| CIFAR10-C | 23.0 | 22.9 | 23.0 | 22.9 |
| CIFAR100-C | 34.6 | 34.7 | 34.8 | 34.8 |
| IN-C (BN) | 69.0 | 67.2 | 67.6 | 67.0 |
| IN-C (LN) | 49.0 | 48.8 | 48.7 | 48.8 |
| **COMM_INTERVAL** | 1 | **10** | 100 | 1000 |
| CIFAR10-C | 22.6 | 22.9 | 22.6 | 22.0 |
| CIFAR100-C | 34.7 | 34.7 | 34.9 | 43.2 |
| IN-C (BN) | 67.1 | 67.2 | 67.2 | 67.4 |
| IN-C (LN) | 48.4 | 48.8 | 48.8 | 48.7 |

For the sensitivity analysis of COMM_INTERVAL, we observe that our method is generally robust to changes in the communication interval across all datasets. However, the impact of communication frequency varies significantly among different datasets. For simpler datasets like CIFAR10, minimal communication, exemplified by an interval of $f = 1000$, yields the best results. This could be attributed to the model's high accuracy, enabling positive feedback loops even within isolated branches. Conversely, for more complex datasets, more frequent communication, with intervals as low as $f = 1$, appears beneficial. This frequent updating may help prevent model degradation over time, especially in scenarios where the data complexity could lead to significant divergences in learning pathways among distributed model components.

## E DATASET VISUALIZATION

To further illustrate the characteristics of the datasets used in our evaluation, we present visualizations of the data distribution across different corruption types (Fig. 3), natural domain shifts (Fig. 4), and medical centers (Fig. 5). These figures highlight the diverse challenges that our models face in each evaluation scenario, providing insight into the complexity of the test conditions.

## F LIMITATION AND FUTURE WORK

While FreDA addresses a critical challenge in handling heterogeneous data streams, providing a solid pipeline for this issue, there are still avenues for further enhancement.

On the theoretical front, although our framework has demonstrated its effectiveness empirically, developing a more formal understanding of its convergence and optimality could further solidify its foundations and provide additional clarity on its broader applicability.

In terms of practical optimization, our current aggregation approach, which averages models based on cluster counts, has been effective in solving the problem at hand. However, exploring alternative strategies—such as weighting models by the divergence between clusters—might lead to incremental improvements. Additionally, refining the sample selection process from a original sample-level focus to a more granular patch-level could extend FreDA's applicability to tasks such as segmentation, thereby further enhancing its versatility in real-world scenarios.

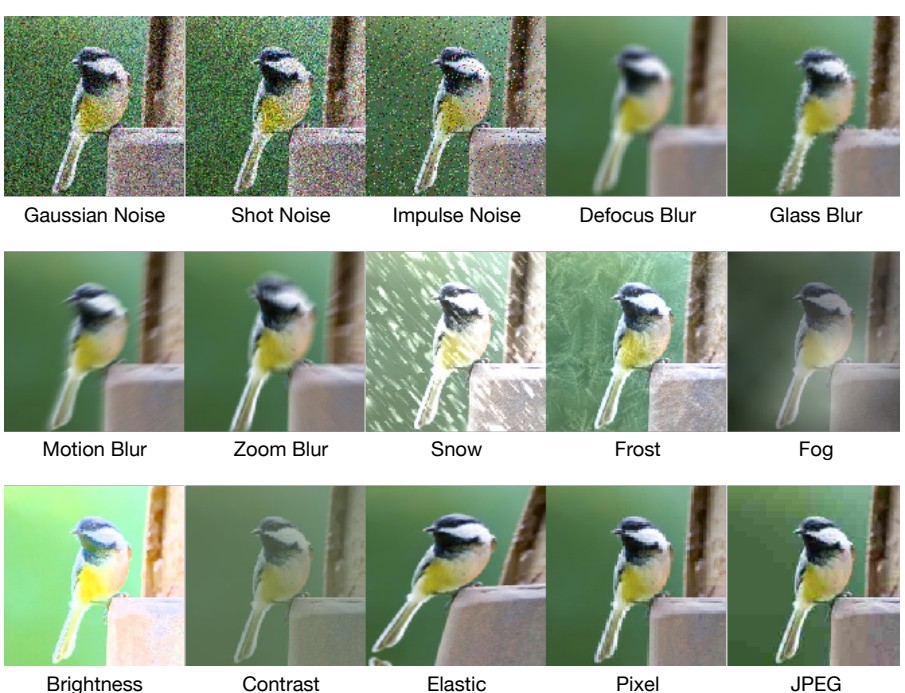

Figure 3: Examples from ImageNet-C under common image corruptions. The images showcase a range of corruption types (e.g., noise, blur, and weather distortions) at varying severity levels.

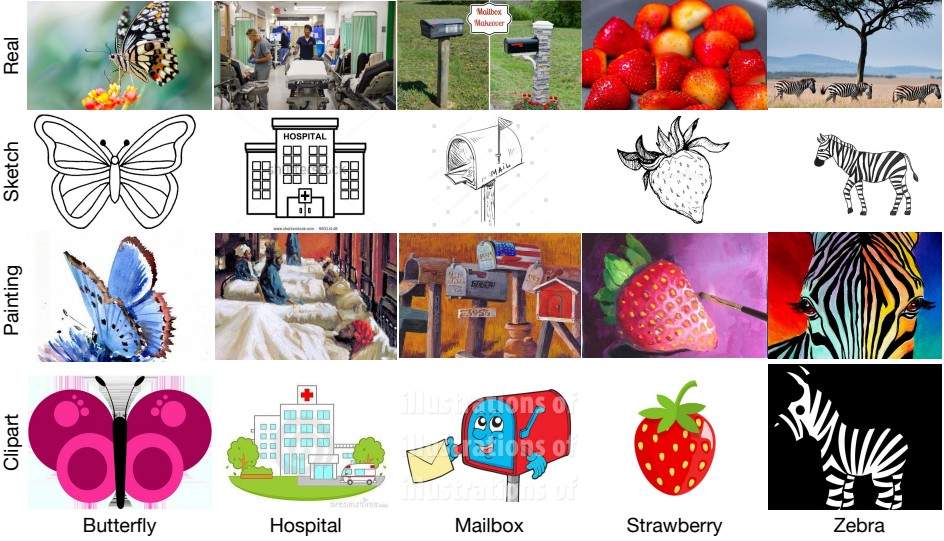

Figure 4: Samples from DomainNet126 across four subdomains (Real, Sketch, Painting, Clipart). These visualizations reflect the stylistic and perceptual variations inherent in each domain.

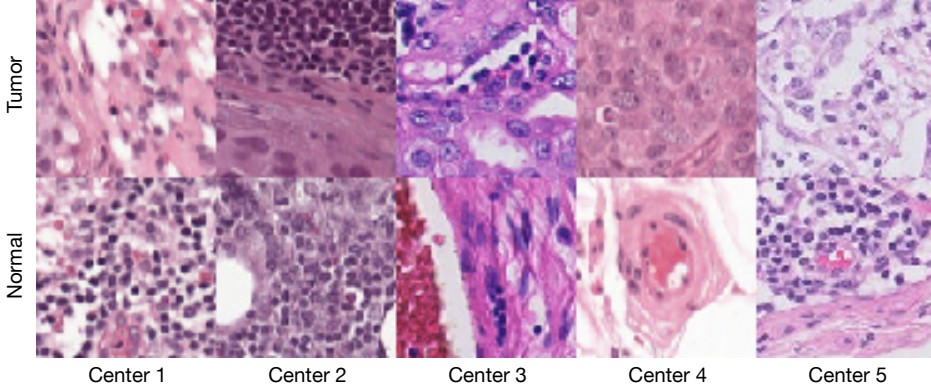

Figure 5: Example patches from the Camelyon17 dataset, containing histopathological images used for tumor detection.

