# OpenReview forum: "Decentralizing Test-time Adaptation under Heterogeneous Data Streams"
_ICLR.cc/2025/Conference — ICLR 2025 Conference Withdrawn Submission_

### Official Review · Reviewer_SEHB · 2024-10-28

**Soundness:** 3
**Presentation:** 2
**Contribution:** 3
**Rating:** 6
**Confidence:** 3

**Summary:**

This paper addresses the limitations of Test-Time Adaptation (TTA) in heterogeneous data streams by proposing a Frequency-based Decentralized Adaptation (FDA) framework. The FDA framework leverages Fourier space decomposition to transition data from globally heterogeneous to locally homogeneous, enabling decentralized adaptation for diverse distribution shifts. It employs multiple local models that independently adapt to specific data segments and share knowledge to create a cohesive global model. Additionally, a Fourier-based augmentation strategy is introduced to enhance model robustness. The proposed approach is shown to outperform existing methods in various domains, including corrupted, natural, and medical settings.

**Strengths:**

1) This investigates a practical and meaningful TTA scenario where it deals with heterogeneous data streams.
2) The data center approach proposed in the article is quite innovative. By leveraging frequency-based decomposition and decentralized adaptation, it offers a reasonable solution to the problem of heterogeneous data streams in Test-Time Adaptation.
3) The proposed method has demonstrated its effectiveness through a considerable number of real-world data experiments.

**Weaknesses:**

1) The method proposed in this work lacks a theoretical analysis of its generalization boundaries and an explanation of its limitations.
2) Paper highlights that its approach is data-centric rather than model-centric. However, since most TTA methods also involve data-based transfer, what does it mean for the paper to be data-centric rather than model-centric?

**Questions:**

Please see weakness.

**Details Of Ethics Concerns:**

nan

---

### Official Review · Reviewer_gyo3 · 2024-10-28

**Soundness:** 3
**Presentation:** 3
**Contribution:** 3
**Rating:** 5
**Confidence:** 5

**Summary:**

This paper focuses on the test-time adaptation for mixed distribution shifts. Specifically, the authors propose to leverage Fourier information to manage diverse data shifts. The experiments on various benchmarks and the medical tasks show the proposal can improve the performance.

**Strengths:**

1) The proposal focuses on the test-time adaptation for mixed distribution shifts. The problem is novel.
2) The proposal exploits a frequency-based decentralized adaptation method that transits data from globally heterogeneous to locally homogeneous in Fourier space. The idea is clear and interesting.

**Weaknesses:**

1) For the experiments, only the mean performance is reported. How many times were the experiments repeated and how about the variance?
2) The mixed data distribution studied in this paper is the most simple one, i.e., the mix of covariate shifts. In more practical scenarios, covariate shift and label shift may occur simultaneously. How to deal with more complexed distribution shifts is a challenging problem.
3) The experiments are all image classification tasks. Nowadays, some pre-trained vision-language models, such as CLIP, have demonstrated impressive results on image classification tasks. How about the performance of these models and can the proposal be combined with these models?
4) Some important related works, such as [1], are not discussed.

[1] ODS: Test-Time Adaptation in the Presence of Open-World Data Shift.  ICML 2023.

**Questions:**

As discussed above.

---

### Official Review · Reviewer_6NX5 · 2024-11-04

**Soundness:** 3
**Presentation:** 3
**Contribution:** 2
**Rating:** 5
**Confidence:** 3

**Summary:**

The paper tackles the challenge of heterogonous data in TTA through the lens of Fourier transfer. Firstly, the author does some analysis and shows that utilizing high-frequency features, which are extracted from the input image through the Fourier operator and then filtering out low-frequent ones, can enhance the discriminative between domain shifts. Based on this observation, the author has clustered the input data into multiple clusters and then adapted the samples in each cluster with a local model. After a few steps, these local models connect to share the information and update their weight. Besides, the author introduces the data augmentation techniques, which add Gaussian noise to the frequency domain to enhance model performance.

**Strengths:**

1. The heterogeneous data in TTA is important and one of the hardest problems in the TTA setting.
2. Strong motivation, the analysis of the discriminative of high-frequence features in Figure 1 seems interesting, The authors then based on it to develop their main algorithm

**Weaknesses:**

1. The main limitation of this paper is its novelty. The most novelty comes from clustering, and training multiple models locally. About the augmented process, augmentation on the frequency domain seems not a new idea [1].
2. Running time: The mechanic that uses multiple local models, combing with Fourier transform, and one more step adapt on augmentation data, has slowed down the updating performance of the whole algorithms.
3. Lack ablation study: The authors's contribution could be split in two-fold: clustering adaptation and Fourier augmentation, so it should be important to understand how these components work alone, especially the first module (clustering adaptation).

Ref:

[1] Fourier-basis Functions to Bridge Augmentation Gap: Rethinking Frequency Augmentation in Image Classification (CVPR2024)

**Questions:**

1. The mechanic updates knowledge of local models: If the reviewer understands correctly, clustering aims to divide data into multiple distributions, and each distribution (cluster) will be adapted by one local model. Then they will weigh this knowledge to achieve the global one, then each local model will be assigned by this global (Equation 6). The reviewer's concern is doing this update loses too much information from the local one because updating in this way can shift the local weight too far from its previous version, which can reduce the adaptation performance on its local domain.

2. Could the author report the running time of your models compared with other baselines?
3. The naturalness of generated images: Could the author provide some generated images when after applying your augmented modules?

---

### Official Review · Reviewer_qX5B · 2024-11-04

**Soundness:** 2
**Presentation:** 2
**Contribution:** 2
**Rating:** 5
**Confidence:** 3

**Summary:**

The paper introduces a data-centric approach to Test-Time Adaptation (TTA) called Frequency-based Decentralized Adaptation (FreDA). The method employs high-frequency Fourier features to cluster similar distribution shifts, enabling decentralized learning where separate models adapt to each cluster. The framework includes a novel Fourier-based augmentation strategy and periodic global parameter aggregation across clusters to maintain overall model coherence. This approach effectively addresses the challenge of heterogeneous data streams in TTA, which previous methods have struggled to handle.

**Strengths:**

- Novel application of Fourier analysis in TTA that enables effective separation of different distribution shifts
- Strong empirical results across multiple datasets and settings, with significant improvements over state-of-the-art methods

**Weaknesses:**

- Limited technical novelty in individual components, e.g., weight aggregation scheme closely resembles existing federated learning approaches, Fourier transformation and filtering techniques are conventional


- Presentation and clarity issues: Inconsistent notation: parameters are denoted as $\theta_k$ in Equation 4 but as $W_k$ in Equation 5 without explanation
Missing important visualizations that can benefit the analysis in paper, e.g. examples of images after low-frequency filtering, visualization of cluster assignments or their characteristics, limited illustration of how different distribution shifts are captured by the Fourier features

**Questions:**

See above

---

### Note · Authors · 2024-11-14

I have read and agree with the venue's withdrawal policy on behalf of myself and my co-authors.